# Rickettsia Vaccine Candidate pVAX1-OmpB24 Stimulates TCD4+INF-γ+ and TCD8+INF-γ+ Lymphocytes in Autologous Co-Culture of Human Cells

**DOI:** 10.3390/vaccines11010173

**Published:** 2023-01-13

**Authors:** Karla Dzul-Rosado, Luis Donis-Maturano, Juan Arias-León, Jesús Machado-Contreras, Guillermo Valencia-Pacheco, Candi Panti-Balam, Javier Balam-Romero, Angela Ku-González, Gaspar Peniche-Lara, Juan Mosqueda, Oscar E. Zazueta, Cesar Lugo-Caballero, Fernando Puerto-Manzano

**Affiliations:** 1Dr. Hideyo Noguchi Regional Research Center, Autonomous University of Yucatan, Mérida 97000, Mexico; 2Faculty of Higher Studies-Iztacala, National Autonomous University of Mexico, Tlalnepantla de Baz 54090, Mexico; 3Faculty of Medicine, Autonomous University of Yucatan, Mérida 97000, Mexico; 4Immunology Laboratory, Faculty of Medicine, Autonomous University of Baja California, Mexicali 21000, Mexico; 5Yucatan Scientific Research Center, Biochemistry and Plant Molecular Biology Unit, Mérida 97200, Mexico; 6Immunology and Vaccines Laboratory, College of Natural Sciences, Autonomous University of Queretaro, Queretaro 76010, Mexico; 7Ministry of Health of Baja California, Mexicali 21000, Mexico

**Keywords:** activated lymphocytes, pVAX1-OmpB24, *Rickettsia rickettsii*

## Abstract

Background: In recent years, promising vaccination strategies against rickettsiosis have been described in experimental animal models and human cells. OmpB is considered an immunodominant antigen that is recognized by T and B cells. The aim of this study was to identify TCD4+INF-γ+ and TCD8+INF-γ+ lymphocytes in an autologous system with macrophages transfected with the vaccine candidate pVAX1-OmpB24. Lymphocytes and monocytes from 14 patients with Rickettsia were isolated from whole blood. Monocytes were differentiated into macrophages and transfected with the plasmid pVAX1-OmpB24 pVax1. Isolated lymphocytes were cultured with transfected macrophages. IFN-γ-producing TCD4+ and TCD8+ lymphocyte subpopulations were identified by flow cytometry, as was the percentage of macrophages expressing CD40+, CD80+, HLA-I and HLA-II. Also, we analyzed the exhausted condition of the T lymphocyte subpopulation by PD1 expression. Macrophages transfected with pVAX1-OmpB24 stimulated TCD4+INF-γ+ cells in healthy subjects and patients infected with *R. typhi*. Macrophages stimulated TCD8+INF-γ+ cells in healthy subjects and patients infected with *R. rickettsii* and *R. felis*. Cells from healthy donors stimulated with OmpB-24 showed a higher percentage of TCD4+PD1+. Cells from patients infected with *R. rickettsii* had a higher percentage of TCD8+PD-1+, and for those infected with *R. typhi* the larger number of cells corresponded to TCD4+PD1+. Human macrophages transfected with pVAX1-OmpB24 activated TCD4+IFN-γ+ and CD8+IFN-γ+ in patients infected with different Rickettsia species. However, PD1 expression played an important role in the inhibition of T lymphocytes with *R. felis.*

## 1. Introduction

Rickettsioses are worldwide febrile infectious diseases caused by small obligate intracellular bacteria of the genera Rickettsia: spotted fever (SFG) and typhus groups (TG) are responsible for rickettsiosis as emerging diseases [1]. In surveys of febrile illness, the proportion of fever attributed to rickettsia varies from 0.9% to 18% [2]. Oral doxycycline is effective as treatment, but in the case of severe life-threatening disease intravenous therapy is recommended. Unfortunately, doxycycline is not always available [3]. 

There have been concerted efforts to develop an effective vaccine against Rickettsia. Rickettsia proteins linked to cellular invasion (invA), division (fts), protein secretion (sec gene family) and virulence (OmpA, OmpB, virB, cap, Adr2, YbgF, TolC, tlyA and tlyC gene family), have been implicated [4,5,6,7]. Different vaccines containing non-viable *R. rickettsii* have been prepared from infected tick tissues [8], eggs [9], and embryonic chicken cells [10]. These vaccines have been tested on humans but have only been proven to be minimally effective in preventing Rocky Mountain Spotted Fever (RMSF) [11,12]. The generation of DNA vaccines against the rOmpA and rOmpB has led to a high immune response in murine models infected with *R. conorii* and *R. rickettsia* [13]. However, evaluation of rOmpB protein in murine models might be a limiting factor when compared with human models [14].

Because of the intracellular nature of *R. rickettsii*, an experimental strategy is to identify T lymphocytes that specifically recognize epitopes presented by HLA-I molecules. Naturally activated lymphocytes can be reactivated in vitro after a second exposure and their immune activity can be compared with that of the lymphocytes from healthy subjects [15]. Human ex vivo trials using macrophages or dendritic cells as antigen-presenting cells (APC), in co-culture with sensitized lymphocytes, have created high expectations, with higher predictive value than any preclinical strategy [16]. In preliminary studies, it has been shown that peripheral blood mononuclear cells (PBMC) on patients previously infected with rickettsia generated proliferative responses after stimulation with pVAX1-OmpB24, with production of IL-2, IFN-γ, IL-12p70, IL-6 and TNF-α in heterologous co-culture supernatants [17]. However, the extent of their recognition and processing by the human immune system and potential HLA restriction is still unclear. Our objective was to identify TCD4+IFN-γ+ and TCD8+IFN-γ+ lymphocytes induced in an autologous system by macrophages transfected with the vaccine candidate rickettsia pVAX1-OmpB24.

## 2. Materials and Methods

### 2.1. Study Population

It was included in the study two groups: one of patients and other one of healthy subjects, all of them needed to be over 18 years of age, born in Mexico, matched for age and sex, with no history of autoimmune, immunosuppressive, or infectious diseases, were included in the study. In the study were excluded patients with any acute or newly vac-cinated infections. Blood samples were obtained from patients and healthy subjects who signed the informed consent. The work was approved by Ethics Committee of Hideyo Noguchi Regional Investigation Research Center (Merida, Yucatan, Mexico) (CIE-05-2017).

### 2.2. Molecular Identification of Rickettsia in Patients 

DNA extraction was performed using a QIAGEN QIAamp DNA kit according to the manufacturer’s instructions. PCR was performed to amplify a region of 17 kDa (htrA) and rOmpB [18,19]. Positive PCR amplicons were sequenced. 

### 2.3. Isolation of Lymphocytes and Monocytes from Peripheral Blood

Blood from patients and controls was collected in heparinized tubes (Vacutainer, BD, Hombrechtikon, Switzerland) for purification of PBMC. The blood was diluted 1:1 with Hank’s balanced saline solution (HBSS) pH 7.4 (5.33 mM KCl, 0.44 mM KH_2_PO_4_, 137.93 mM NaCl, 0.33 mM Na_2_HPO_4_), then it was added to a Ficoll-histopaque solution at a ratio of 3:1 and centrifuged at 574× *g* max. RCF for 20 min at 4 °C. The separated blood had a ring of white blood cells between the ficoll phase and the plasma. The leukocytes were taken and placed into a 15 mL Falcon tube. 14 mL of PBS 1X was added to the Falcon tube and it was then centrifuged at 574× *g* max. RCF for 8 min at 24 °C for the first wash. After that, the supernatant was removed. 14 mL of PBS 1X was added again to the supernatant and it was then centrifuged at 380× *g* max. RCF for 8 min at 4 °C. This wash was repeated 2 times. One mL of sterile distilled water was added and combined with 14 mL of PBS 1X. It was again centrifuged at 380× *g* max. RCF for 8 min at 4 °C. The ring of PBMC was collected and washed twice with HBSS at pH 7.4 and then put in a complete medium [RPMI-1640 medium (Gibco, Grand Island, NE, USA), with 1000 μg/mL streptomycin, 2 mM L-glutamine and 10% fetal bovine serum (FBS)].

Monocyte purification was achieved by density centrifugation using hyperosmotic Percoll Plus® at a density adjusted to 1.064 g/mL and 1.05 g/mL. A 10 mL aliquot of Percoll® solution was added to a 15-mL centrifuge tube and the PBMC suspension layered on the Percoll® solution in 200 µL aliquots at a speed of approximately 200 µL/60 s using a gel loading tip of 1–200 µL. The preparation was centrifuged at 169× *g* max. RCF for 15 min at 25 °C; brake and the monocytes within a band located at the interface of the IMDM (Iscove’s Modified Dulbecco’s Medium) solution and Percoll® were collected and diluted with 4 mL of HBSS. The lymphocyte ring was cultured in cell culture flasks in a humidified incubator with 5% CO_2_ at 36 °C. The cells were sedimented by centrifugation at 169× *g* max. RCF for 10 min at 25 °C and the supernatant discarded. Monocytes were resuspended in 1 mL of IMDM, and the number of cells determined using the trypan blue exclusion method and Neubauer chamber count. All cell suspensions of T cells were stained with trypan blue to be counted and to assess their viability. In this way, all the trials had a feasibility greater than 95%.

### 2.4. Differentiation and Transfection of Monocyte-Derived Macrophages

Monocytes were seeded on 24-well culture plates (Costar, Corning, NY, USA) at a density of 3 × 10^5^ cells/well and a final volume 600 μL/well. The plates were transferred to a humidified incubator with 5% CO_2_ at 36 °C for a period of 2 h. Afterwards, the supernatant containing non-adherent cells was removed by a gentle rinse and the medium replaced with IMDM containing 100 ng/mL of GM-CSF. This was left for 6 days, with medium changes being made on days 1 and 4, removing the supernatant by a gentle rinsing of the cell, until finally macrophages were obtained. 

*Transfection using lipofectamine.* Lipofectamine (LTX, 5 μL) and Plus reagent (5 μL, Opti-MEM, Invitrogen) were added to plasmid pVAX1-OmpB24 (1 μL) previously obtained [17]. The mixtures were incubated at 24 °C for 10 min to allow complexes of DNA-LTX, which were added directly to the macrophages (100 μL/well) for 24 h at 37 °C and 5% CO_2_. Empty plasmid was used as a negative control. To verify transfection, RNA extraction was carried out on transfected macrophages, recovery being by mechanical action: scraping and centrifugation after 12, 24, 48 and 78 h, using TRIzol reagent. Purified RNA was resuspended in DEPC water, quantified by spectrophotometry, and stored at −70 °C until use. The cDNA was obtained by RT-PCR using the ImProm II reverse transcriptase system kit (Promega, Madison, WI, USA) cunder the following conditions: 25 °C for 5 min for alignment, 42 °C for 60 min, and inactivation at 70 °C for 15 min, followed by 94 °C for 30 s for denaturation, 30 s at 62 °C for annealing and 3 min at 72 °C for extension, for 35 cycles. The primers used to amplify the OmpB-24 sequence were: forward 5-ATGGTCGGTGGACAGCAAGGTAATAAG-3 and reverse 5-GTAACGATAGCTCCAACAAAG-3. 

Transfection was also validated by indirect immunofluorescence (IFI). For this, the transfected macrophages were fixed on glass coverslips, and detection of the expressed antigens was carried out using immune serum and preimmunized serum diluted 1:64 with PBS 1x, and 3% skimmed milk powder. The coverslips were incubated for 1 h at 37 °C in 5% CO_2_ and 2 washes were performed with 1 mL of PBS-0.05% Tween-20 for 5 min at room temperature. Subsequently, the FITC-conjugated anti-human IgG secondary antibody (Jakson Immuno R®, West Grove, PA, USA), was diluted in PBS 1X with 3% skimmed milk powder, and the anti-DAPI (CTR® scientific, Monterrey, Nuevo Leon, Mexico) antibodies were added, and incubated at 37 °C for 1 h in 5% CO_2_. A final wash was performed and incubated for 5 min at room temperature. Finally, the coverslips were removed from the culture box, allowed to dry, and drops of glycerol were added before they were sealed for observation under the confocal microscope, image was observed at 40× (LSMC FV-1000, Olympus, Tokyo, Japan). 

### 2.5. Autologous Culture of Lymphocytes with Transfected Macrophages

Lymphocytes recovered by centrifugation gradient were cultured with transfected macrophages (1.5 × 10^5^ per well at 2:1 ratio) in an RPMI-1640 medium free of antibiotics, for 72 h at 37 °C in 5% CO_2_. Triplicate assays were performed using lymphocytes and non-transfected macrophages: lymphocytes from sensitized subjects with transfected macrophages, lymphocytes with macrophages transfected with empty pVAX1 plasmid as a negative control, and macrophages and lymphocytes stimulated with concanavalin (ConA) as a positive control. All cell suspensions of T cells were stained with trypan blue to be counted and to assess their viability. In this way, all the trials had a feasibility greater than 95%.

### 2.6. Flow Cytometric Analysis

Using the FACSverse flow cytometer (Becton Dickinson), we proceeded to identify the populations of LcT CD4+INF-γ+ and LcT CD8+IFN-γ+, as well as the antigenic presentation of transfected macrophages using the FlowJo 10.8.1 program and GraphPad 8.0.1 to process data obtained in the analysis and construction of graphs. Surface immunostaining was performed on the lymphocytes using monoclonal antibodies: anti-human CD3-PE, anti-human CD4-APC and anti-human CD8-PercP. Then, an intracellular staining using the monoclonal anti-human INF-γ-Alexa 488 was done. Antigenic presentation in macrophages was analyzed by expression of costimulatory molecules and MHC-I/MHC-II, with the following monoclonal antibodies: anti-human CD14-FITC, anti-human CD40-PE, anti-human CD80-APC, anti-human HLA-I-Pacific Blue, and anti-human HLA-II-PerCP. Additionally, the level of expression of PD-1 in T lymphocytes was determine, by means of the anti-human PD1 antibody, to establish the degree of cellular fatigue. All antibodies were from BD (San Jose, CA, USA). Mean Fluorescence Intensity (MFI) of costimulatory molecules and MHC-I/MHC-II was evaluated in the patient and control groups.

## 3. Results

### 3.1. Study Population 

A total of 14 patients, 9 from Yucatan and 5 from Baja California, Mexico, were positive for *R. felis*, *R. typhi*, and *R. rickettsii*, confirmed by PCR. The healthy subjects were of the same age, region, and ancestry. None of the patients or healthy subjects were under treatment prior to enrollment in the study (Table 1).

### 3.2. Transfection of Macrophages with pVAX1-OmpB24

The expression of the mRNA from OmpB-24 in transfected macrophages was verified 12 to 72 h post-transfection by real time PCR, in which the expected 1.7 kb amplicon was detected. We chose 24 h as the standard time for the following experiments with the exogenous protein. (Figure 1).

Transfection efficiency was assessed by indirect immunofluorescence (IFI) using a previously generated mouse antiserum against full-length recombinant OmpB-24 and a FITC-labeled anti-mouse secondary antibody. Viable transfected cells were identified by confocal microscopy using DAPI counterstaining and observing at least 5 different fields. The mean number of viable transfected cells was 90.86% vs. 0% for non-transfected cells (Figure 2), which was corroborated by fluorescence-activated cell sorting (FACS) using indirect staining against OmpB-24. (Data not shown).

### 3.3. Flow Cytometric Analysis

#### TCD4+INF-γ+ and TCD8+INF-γ+ Lymphocytes Induced by Human Macrophages Transfected with pVAX1-OmpB24

In the co-cultures of cells stimulated with pVAX1-OmpB24, from healthy donors and patients infected with *R. rickettsii*, higher percentages of TCD8+INF-γ+ were observed: 11.74% and 10.3%, respectively (Figure 3a). Meanwhile, the co-cultures of cells from healthy donors and patients infected with *R. typhi*, showed increases in TCD4+INF-γ+ of 15.29% and 8.6%, respectively, with significant differences between the two groups (Figure 3b). In the case of co-cultures of cells from healthy donors and patients infected with *R. felis*, figures of 23.5% and 30.2% respectively were recorded for the population of TCD8+INF-γ+ (Figure 3c).

When determining the antigenic presentation of transfected macrophages, higher percentages of CD40+, CD80+, HLA-I and HLA-II were observed in macrophages from healthy donors transfected with the OmpB-24 plasmid compared to those from patients infected with *R. felis, R. typhi,* and *R. rickettsii*. (Table 2). However, when evaluating the MFI (Mediam Fluorescence Intensity) in the group of patients with *R. felis*, a higher expression of CD40+, CD80+, HLA-I and HLA-II was observed compared with the controls (Table 3). Macrophages from patients stimulated with OmpB-24 showed a low percentage compared to healthy subjects, but had a higher IMF. 

The results showed a higher percentage of CD14+CD80+ cells in the co-cultures of healthy donors and patients infected with *R. rickettsii*, with OmpB-24 stimulation and an increase in the CD14+HLA-I population. In contrast, in the co-cultures of healthy donors and patients infected with *R. typhi*, an increase in the percentage of CD14+CD40+ was observed, while the CD14+HLA-I population was increased with the stimulation of OmpB-24. In the co-cultures of healthy donors and patients infected with *R. felis*, an increase occured in the CD14+HLA-I population, compared to the CD14+HLA-II population. However, the difference between the two populations was minimal (Table 2).

The FMI results showed higher CD80 expression in CD14+ macrophages of healthy donors and patients infected with *R. rickettsii* compared to OmpB-24 stimulation, and higher expression of the HLA-I. On the other hand, in macrophages from healthy donors and patients infected with *R. typhi*, greater expression of CD40 was observed, while the expression of HLA-I was increased with OmpB-24 stimulation. In macrophages from healthy donors and patients infected with *R. felis*, an increased expression of CD80 and HLA-I was observed, and the expression of HLA-II was similar to HLA-I in the patient group (Table 3).

### 3.4. PD1 Expression in TCD4+ and TCD8+ Lymphocytes

In co-cultures of cells stimulated with OmpB-24 from healthy donors, a greater population of TCD4+PD1+ was observed, with 35.94%, while in patients with *R. rickettsii* the highest percentage was found in TCD8+PD-1+, with 6.34% (Figure 4a). Cells from healthy donors and patients with *R. typhi* under the stimulation of OmpB-24 showed increases in TCD4+PD1+ of 11.85% and 10.1%, respectively (Figure 4b). In the case of healthy donors and patients with *R. felis*, there was an increase in the response of TCD8+PD1+. However, the difference in percentages was minimal compared to TCD4+PD1+ (Figure 4c). Interestingly, TCD4+PD1+ and TCD8+PD1+ cells from the group of patients with *R. rickettsii* and *R. typhi* in co-cultures with ConA showed an increase in the production of INF-γ.

## 4. Discussion

The social impact of the development of a vaccine against *Rickettsia* is important since there are currently no commercial methods for the diagnosis of rickettsiosis in the acute phase, which is found mainly in vulnerable communities away from specialized health services where patients are in critical conditions that eventually become fatal cases [20]. Furthermore, the initial clinical considerations are like other pathologies such as Dengue or chikungunya which delays the diagnosis and treatment [21].

The nature of rickettsial antigens that elicit humoral and cellular responses has been the subject of thorough investigations, primarily directed toward vaccine development. However, the immunodominant rickettsial antigens responsible for stimulating CD4+ and CD8+ cells have not yet been clearly identified and existing studies have been based on animal models. In recent years, *ex vivo* assays have been done using presenting cells (macrophages or DC) maintained in cultures with sensitized lymphocytes, and the determination of the cytokine response generated under this strategy has created high expectations, with high predictive value and a human profile, resulting in potential antigens to be used in the development of human vaccines [22,23]. The use of autologous systems has the advantage of matching the main HLA loci for cases in which antigenic recognition is required. In addition, it allows for the anticipated selection of cell clones. In the present study, an autologous system was used with macrophages transfected with the vaccine candidate Rickettsia pVAX1-OmpB24 [24,25,26].

Macrophage transfection by cationic liposomes is a highly effective technique which guarantees efficient delivery of DNA plasmids into cells. Transfection efficiency has been evaluated by methods such as fluorescent microscopy, gene reporter activity, PCR, flow cytometry or viability assays [27,28,29]. PCR, confocal microscopy, and flow cytometry were used to evaluate the transfection efficiency of this plasmid. Confocal microscopy and flow cytometry reported 90.8% of viable cells, underscoring a low percentage of cytotoxic damage to transfected cells. It is known that DNA liposome transfection enhances the generation of specific TCD8 lymphocytes through MHC-I presentation; however, there are several reports of TCD4 lymphocyte-oriented immune responses with these delivery systems [30,31]. It has been hypothesized that a common epitope for MHC I or II molecules could elicit a TCD4 or CD8 response, and that liposome transfection could trigger a cross presentation through both MHC molecules [32].

After antigenic contact, T lymphocytes can have different effector responses. IFN-γ is a cytokine produced mainly by CD4+ lymphocytes, NK cells and macrophages, and it is important for the control or elimination of intracellular pathogens [33]. In heterologous models, levels of IL-2 and IFN-γ related to CD4+ lymphocytes have been detected [17]. The present study shows an approach to the immune response generated by the plasmid pVAX1-OmpB24, using an autologous system of human cells. It was observed that CD4+IFN-γ+ lymphocytes are found in a higher percentage than CD8+IFN-γ+ lymphocytes, which suggests that the plasmid could be inducing a cellular immune response, similar to that obtained in murine models [14]. The presence of CD4+IFN-γ+ and CD8+IFN-γ+ lymphocytes are related to the cellular response and the elimination of intracellular pathogens. However, patients presented a different CD4+IFN-γ+ expression from that of healthy subjects, and a higher rate of CD4+ lymphocyte proliferation. Variables such as the age of the subjects can influence the immune response. Increased age is linked to immunosenescence, which translates to a decrease in virgin cells compared to existing memory cells [34]. This data shows that during a first infection the immune response is oriented towards cellular infection, for the elimination of the pathogen through mechanisms dependent on IFN-γ [1].

The immune system responds to rickettsial infection. However, there is a decrease in the percentage of CD4+ and CD45+ lymphocytes, as well as an increase in monocytes, which maintain high levels of TNF-α [35]. IFN-γ and TNF-α are cytokines that activate phagocytic cells such as macrophages, and the release of IL-12 influences the presentation of antigens to CD8+lymphocytes, inducing the destruction of cells infected by bacteria [36]. Meanwhile, the presence of IL-4 and IFN-γ favosr antigen presentation to CD4+ lymphocytes. It has been reported that infected endothelial cells can carry out autophagy and promote antigen presentation to B or CD4+ lymphocytes mediated by MHC-II [37,38].

After contact with antigens, T lymphocytes can have activating or inhibitory responses. PD1 (Programmed dead-1) is a molecule that inhibits some lymphocyte functions such as cytokine production, cell proliferation and cytotoxicity. The interaction of PD1 with its ligands PD-L1 and PD-L2 is important in the control of T lymphocyte activation in infectious diseases, since it limits the response of subpopulations of effector cells to avoid tissue damage [39]. The expression level of PD1 in T lymphocytes is a useful marker to establish the degree of cellular exhaustion, which seems to correlate with chronic infections [40]. It has been reported in *Mycobacterium tuberculosis* infections that PD1 has an important role in inhibiting the cellular response [41,42]. Cellular exhaustion could explain the results found in autologous co-cultures of healthy subjects and patients (post-*R. rickettsii* infection). In homologous co-cultures of healthy subjects and patients (post-*R. felis* infection) a negative regulation by PD1 was not seen, possibly explaining the low degree of pathogenicity that is observed in the clinical evolution of these patients. The result observed in co-cultures of healthy subjects and patients (post-*R. typhi* infection), could be associated with other molecules such as CTLA-4, as it occurs in infections with *Leishmania donovani* and *Mycobacteria* However, studies are required to prove this [43,44]. This preliminary data will allow further studies to determine the role of PD1 in *Rickettsia* infections.

As for the molecules related to antigenic presentation in macrophages, low expression of HLA-I in patients was observed. This could be due to the degradation of the plasmid containing pVAX1-OmpB24, which prevents its being processed by the cytosolic pathway, coupled with the time at which co-cultures were analyzed. It is important to consider these aspects and perform the expression kinetics of this molecule in subsequent studies. The high percentage of HLA-I expression in healthy subjects seems to indicate processing of the rickettsial protein by the cytosolic pathway in the first interaction with *Rickettsia*. No statistical differences were found in antigen presentation by HLA-II expression, probably due to the limited number of patients in the study [17,45,46,47]. The high expression levels of CD80 in macrophages of healthy subjects seems to correlate with the activation of lymphocytes, due to the interaction of CD40L/CD40, and an increase in cytokines such as IL-12 and IFN-γ that promote their differentiation [48]. The high expression of CD80 becomes detectable 24 h after activation, reaching maximum levels between 48 and 72 h. However, it will be necessary to perform an expression kinetics on this molecule in further studies [45,46,47,49]. In macrophages of healthy subjects, there seems to be a greater antigenic presentation mediated by HLA I molecules that allow lymphocytes to interact and express CD80 as a second activation signal, since in the absence of stimulation its expression is low or null [50]. Interaction between MHC-peptide-TLRs, co-stimulatory molecules (CD80 with its ligand CD28) or cytokines provides intracellular signals that allow the stimulation and differentiation of effector cells [50,51]. 

The differences found in the autologous co-cultures of patients who became ill with *R. rickettsii*, *R. typhi* and *R. felis* could be due to the pathogenicity of the species and the genetic diversity of the individuals. Knowledge regarding these immune responses is accumulating, and efforts have been undertaken to identify antigenic components of *Rickettsiae* that may be useful as a vaccine [1,3,52,53,54,55]. 

There are limitations in this study due to the sample size. Although the immune response shows an important behavior, the sample size prevents extrapolation to a larger population.

## 5. Conclusions

The results indicate that for the first time an autologous evaluation model from PBMC can be proposed as an alternative to the use of murine models. This data also suggests cross-reaction of T cells to conserved epitopes of OmpB from SFG and TG *Rickettsiae,* which makes this protein a promising vaccine candidate for vaccination against a broad range of *Rickettsiae*. Activation of TCD4+IFN-γ+ and TCD8+IFN-γ+ lymphocytes has been demonstrated in co-cultures of cells from patients infected with *R. rickettsii*, *R. typhi* and *R. felis* species, and in co-cell cultures from healthy donors, stimulated with the *Rickettsia* vaccine candidate pVAX1-OmpB24, respectively, together with the involvement of PD1 expression in the inhibition of TCD4+ and TCD8+ lymphocytes after Rickettsia infections

## Figures and Tables

**Figure 1 vaccines-11-00173-f001:**
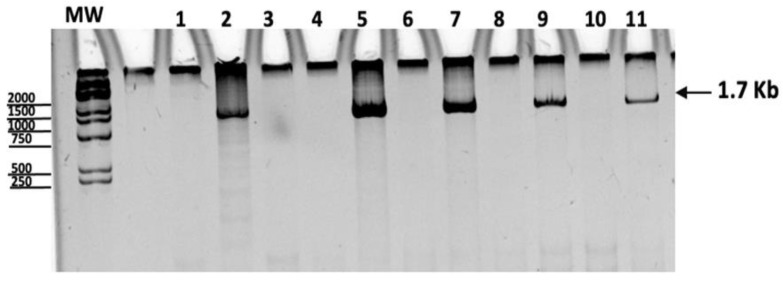
Amplification products by quantitative PCR of macrophages transfected with the plasmid at 24 h. Lane 1: negative control, Lane 2: positive control (pVAX-rOmpB-24 plasmid), Lane 3: non-transfected macrophages. Lanes 4 and 5: Macrophages 12 h after transfection with pVax (4) or OmpB-24 (5), Lanes 6 and 7: Macrophages 24 h after transfection with pVax (6) or OmpB-24 (7), Lanes 8 and 9: Macrophages 48 h after transfection with pVax (8) or OmpB-24 (9), Lanes 10 and 11: Macrophages 72 h after transfection with pVax (10) or OmpB-24 (11).

**Figure 2 vaccines-11-00173-f002:**
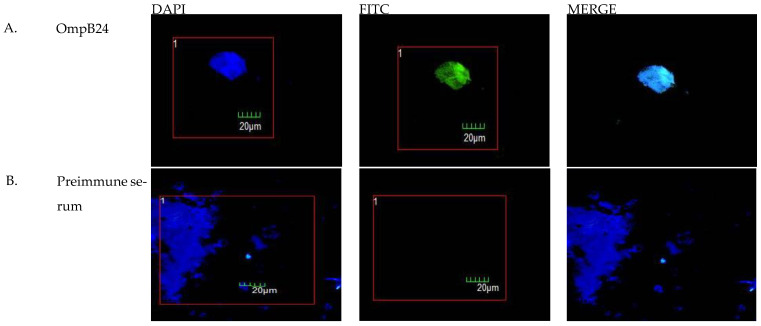
Expression of OmpB-24 in macrophages transfected with pVaX-rOmpB-24. Representative images obtained by confocal microscopy of a single macrophage (20 µm) 24 h after transfection are shown, including (**A**) OmpB-24 immune serum and (**B**) Preimmune serum. Staining for viability (DAPI) and OmpB-24 expression (FITC) are shown in the merge row. image was observed at 40× (LSMC FV-1000, Olympus, Tokyo, Japan).

**Figure 3 vaccines-11-00173-f003:**
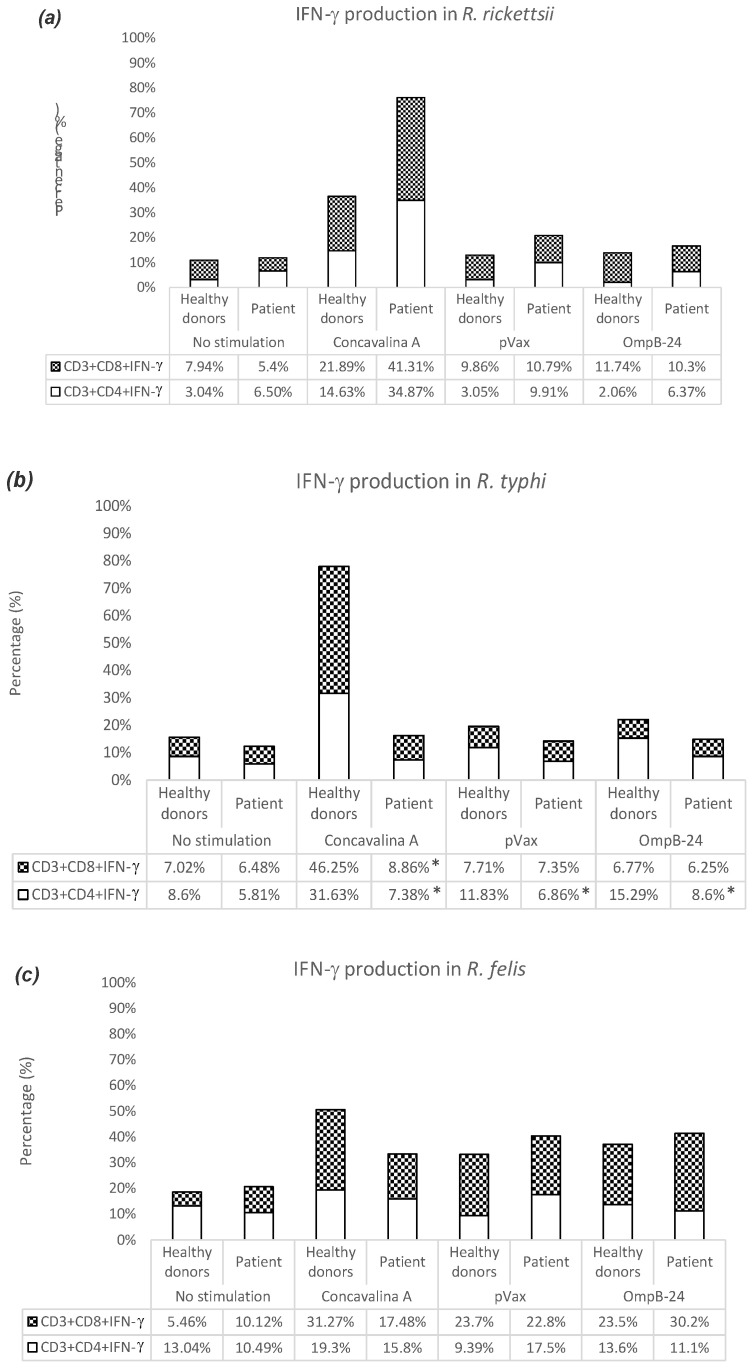
Population of T lymphocytes producing INF-γ. Percentage of TCD4+INF-γ+ and TCD8+INF-γ+ in healthy donors and patients post-infection with *R. rickettsii* (**a**), *R. typhi* (**b**) and *R. felis* (**c**), stimulated or not with ConA, pVax and OmpB -24 for 24 h. Values show the average of 5 samples. (*) *p*-value ≤ 0.05 calculated with the Student’s *t*-test.

**Figure 4 vaccines-11-00173-f004:**
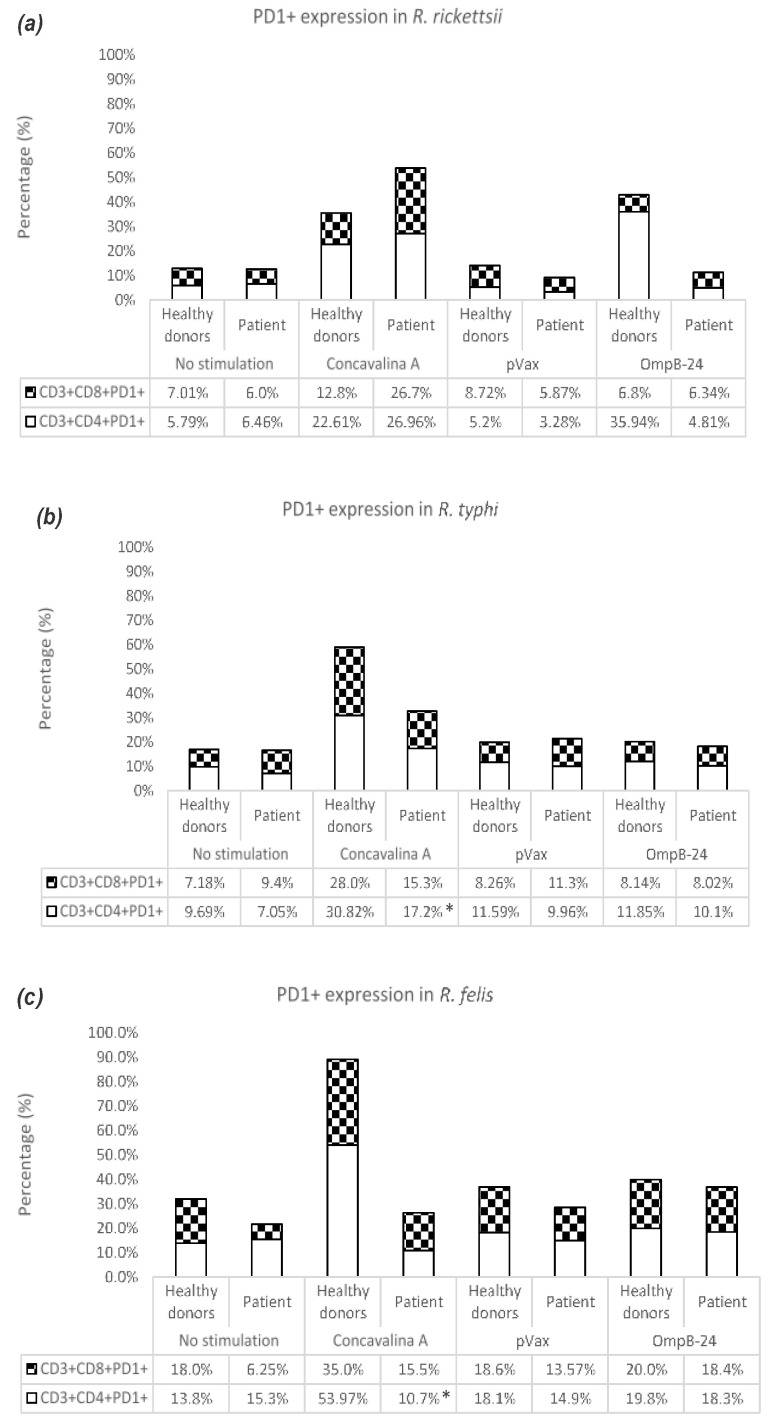
PD1 expression in T lymphocyte subpopulations. Percentage of TCD4+PD1+ and TCD8+PD1+ cells in healthy donors and patients post-infection with *R. rickettsii* (**a**), *R. typhi* (**b**) and *R. felis* (**c**), stimulated or not with ConA, pVax and OmpB- 24 for 24 h. Values show the average of 5 samples. (*) *p*-value ≤ 0.05 calculated with the Student’s *t*-test.

**Table 1 vaccines-11-00173-t001:** Clinical characteristics of the patients.

Age	Gender	State	Symptom Onset Date	Clinical Signs					
				Species	Fever	Exanthema	Headache	Arthralgias/Myalgias	Exposure
**26**	F	Baja California	24/11/16	*R. rickettsii*	+	-	+	+	Tick
**29**	M	Baja California	31/08/16	*R. rickettsii*	+	-	+	+	-
**54**	M	Baja California	18/08/16	*R. rickettsii*	+	-	+	+	Tick
**25**	M	Baja California	-	*R. rickettsii*	+	+	+	+	Tick
**51**	F	Baja California	21/03/17	*R. rickettsii*	+	+	+	+	-
**33**	F	Yucatan	03/10/15	*R. typhi*	+	-	+	+	Fleas
**31**	F	Yucatan	09/11/16	*R. typhi*	+	+	+	-	
**39**	M	Yucatan	14/10/16	*R. typhi*	+	-	+	+	Fleas and Ticks
**28**	M	Yucatan	-	*R. typhi*	+	-	-	-	Tick
**32**	M	Yucatan	-	*R. typhi*	+	-	-	-	-
**72**	M	Yucatan	01/08/13	*R. felis*	+	+	-	+	Tick
**43**	M	Yucatan	01/04/13	*R. felis*	+	-	+	+	Tick
**18**	F	Yucatan	01/05/06	*R. felis*	+	-	-	-	Tick
**16**	F	Yucatan	01/11/13	*R. felis*	+	+	-	-	Tick

**Table 2 vaccines-11-00173-t002:** Percentage of macrophages (CD14+) expressing CD40+, CD80+, HLA-I and HLA-II. $\overset{-}{x}$ refers to the statistical mean.

Macrophage Markers (%)		No Stimulation		Stimulation					
				Concavalina A		pVax		OmpB-24	
		$\overset{-}{x}$							
		Healthy Donors	Patient	Healthy Donors	Patient	Healthy Donors	Patient	Healthy Donors	Patient
** *R. rickettsii* **	CD14^+^CD40^+^	25.30	32.00 ^a^	52.80	11.92	15.20	17.70 ^a^	9.97	12.50 ^a^
	CD14^+^CD80^+^	32.00	23.90	39.40	18.40	27.80	48.10 ^a^	37.30	22.40
	CD14^+^HLA-I	97.80	97.00	95.40	97.00 ^a^	99.20	94.10	98.10	85.50
	CD14^+^HLA-II	46.10	34.30	79.20	21.90	37.90	28.10	37.30	26.60
** *R. typhi* **	CD14^+^CD40^+^	32.70	22.10	15.40	32.20 ^a^	32.90	20.50	39.00	18.30
	CD14^+^CD80^+^	16.70	16.30	11.50	21.00 ^a^	23.00	19.10	22.80	9.91
	CD14^+^HLA-I	99.20	91.80	93.10	96.40 ^a^	93.70	92.20	97.00	97.30 ^a^
	CD14^+^HLA-II	48.00	25.20	27.70	36.10 ^a^	40.10	28.30	53.20	33.50
** *R. felis* **	CD14^+^CD40^+^	10.40	6.70	23.70	4.70	18.90	5.20	18.50	7.30
	CD14^+^CD80^+^	16.60	3.20	27.70	9.20	30.40	9.20	25.90	9.40
	CD14^+^HLA-I	98.90	96.30	97.00	96.80	97.40	97.70 ^a^	97.20	97.70 ^a^
	CD14^+^HLA-II	28.90	66.80 ^a^	33.00	62.10 ^a^	44.60	92.50 ^a^	49.20	94.60 ^a^

(^a^) Higher value observed in patients when compared to their respective control.

**Table 3 vaccines-11-00173-t003:** Mean Fluorescence Intensity (MFI) of CD40, CD80, MHC-I, and MHC-II in macrophages (CD14+). $\overset{-}{x}$ refers to the statistical mean.

(MFI)		No Stimulation		Stimulation					
				Concanavalin A		pVax		OmpB-24	
		$\overset{-}{x}$							
		Healthy Donors	Patient	Healthy Donors	Patient	Healthy Donors	Patient	Healthy Donors	Patient
** *R. rickettsii* **	CD14^+^CD40^+^	4662	2514	4791	2975	4008	3525	4537	2871
	CD14^+^CD80^+^	20,569	2476	29,801	2619	41,331	2275	29,801	3332
	CD14^+^HLA-I	51,054	53,824 ^a^	62,133	65,078 ^a^	58,748	55,552	38,630	42,342 ^a^
	CD14^+^HLA-II	10,110	8117	5039	2000	6000	12,000 ^a^	10,000	8000
** *R. typhi* **	CD14^+^CD40^+^	1270	5136 ^a^	2115	3991 ^a^	1633	4924 ^a^	5191	5439 ^a^
	CD14^+^CD80^+^	3132	2771	1839	4316 ^a^	2400	4352 ^a^	1946	4325 ^a^
	CD14^+^HLA-I	80,092	55,417	34,830	65,897 ^a^	77,776	76,401	82,276	73,816
	CD14^+^HLA-II	7697	9795 ^a^	3325	10,248 ^a^	6972	11,197 ^a^	8010	14,788 ^a^
** *R. felis* **	CD14^+^CD40^+^	5125	189,210 ^a^	2799	465,000 ^a^	6531	340,706 ^a^	147,873	321,110 ^a^
	CD14^+^CD80^+^	2223	47,918 ^a^	2270	49,923 ^a^	2332	95,271 ^a^	2917	125,643 ^a^
	CD14^+^HLA-I	47,429	75,097 ^a^	51,132	65,897 ^a^	62,588	165,240 ^a^	58,606	74,065 ^a^
	CD14^+^HLA-II	9156	72,806 ^a^	7646	78,536 ^a^	12,698	76,929 ^a^	15,171	78,807 ^a^

(^a^) Higher value observed in patients when compared to their respective control.

## Data Availability

The datasets generated and analysed during the present study are available from the corresponding author upon reasonable request.

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
