# Peer review of "Rickettsia Vaccine Candidate pVAX1-OmpB24 Stimulates TCD4+INF-γ+ and TCD8+INF-γ+ Lymphocytes in Autologous Co-Culture of Human Cells"

_vaccines, 2023, doi:10.3390/vaccines11010173_

Round 1

Reviewer 1 Report

Dear respected editor;

Regarding the article number Vaccines-2131023, titled; Rickettsia vaccine candidate pVAX1-OmpB24 stimulates TCD4+INF-γ+ and TCD8+INF-γ+ lymphocytes in autologous co-culture of human cells

This study aimed to identify TCD4+INF-γ+ and TCD8+INF-γ+ lymphocytes in an autologous system with macrophages transfected with the vaccine candidate pVAX1-OmpB24. For this purpose, lymphocytes and monocytes were isolated from the whole blood of 14 patients infected with Rickettsia.

General points to be considered

-          Title: good  

-          Abstract: well written

- This research is well-designed and written.

Many grammatical, typing errors and spaces (ex, lines 92, 94 and 96) should be corrected.

1.     Introduction:

Well written.

2. Materials and Methods

2.1 Study population

Patients and healthy individuals, over 18 years of age, born in Mexico, matched for age and sex, with no history of autoimmune, immunosuppressive, or infectious diseases, were included in the study. Healthy subjects or patients with any acute or newly vaccinated infections were excluded.

-          Inclusion and exclusion criteria should be determined correctly. Healthy subjects are mentioned in both inclusion and exclusion criteria, so please illustrate the difference.

2.5 Autologous culture of lymphocytes with transfected macrophages

Lymphocytes recovered by centrifugation gradient were cultured with transfected 149 macrophages (1.5 × 105 per well at 2: 1 ratio) in RPMI-1640 medium free of antibiotics, for 150 72 h at 37 °C in 5% CO2. Please correct it, it should be superscript.

2.4 Differentiation and transfection of monocyte-derived macrophages 113

Lines 140 and 141; Subsequently, the FITC-conjugated anti-human IgG secondary antibody (Jakson Immuno R® ), diluted in 1………….PBS with 3% skimmed milk powder, missing unit must be added.

Writing units and numbers should be standardized in the whole article. Some different styles can be found at line 95 (4°C) and line 108 (t 36 °C).

3.      Results

Table 1, titled Clinical characteristics of the patients could be improved by minimizing or removing the spaces between the rows.

3.1 Study population

A total of 14 patients (6 females and 8 males), 9 from Yucatan and 5 from Baja California, Mexico, were positive for R. felis, R. typhi, and R. rickettsii, confirmed by PCR. The healthy subjects (how many) were of the same age, region, and ancestry. None of the patients and healthy subjects were under treatment prior to study enrollment (Table 1).

-          Author should mention the number of healthy samples used in this study. Also should mention the numbers of males and females in both patients and healthy samples.

-          Also authors have to compare the differences in the results between males and females.

-          As well as the differences in the percentage of macrophages (CD14+) expressing CD40+, CD80+, HLA-I and HLA-II in both males and females should be statistically compared.

4.      Discussion

-          The first sentence is too long and should be divided and rewritten correctly with references.

The social impact of the development of a vaccine against Rickettsia is important since there are currently no commercial methods for the diagnosis of rickettsiosis in the acute phase, which is found mainly in vulnerable communities away from specialized health services where patients are in critical conditions and, if they don’t find access to treatments, eventually become fatal cases (………..???).

-          At the end of the discussion part, authors have to add the limitation of this study as small sample size and so on……

5.      Conclusion: could be improved.

References:

Well written according to the journal instructions.

Need revision and should be standardized, I have seen some references without page numbers, please recheck all the references. For example; reference 7, and 15.

Author Response

Response to the Reviewer

  1. Materials and Methods

2.1 Study population

Patients and healthy individuals, over 18 years of age, born in Mexico, matched for age and sex, with no history of autoimmune, immunosuppressive, or infectious diseases, were included in the study. Healthy subjects or patients with any acute or newly vaccinated infections were excluded.

-          Inclusion and exclusion criteria should be determined correctly. Healthy subjects are mentioned in both inclusion and exclusion criteria, so please illustrate the difference.

These observations have already been addressed in the manuscript.

2.5 Autologous culture of lymphocytes with transfected macrophages

Lymphocytes recovered by centrifugation gradient were cultured with transfected 149 macrophages (1.5 × 105 per well at 2: 1 ratio) in RPMI-1640 medium free of antibiotics, for 150 72 h at 37 °C in 5% CO2. Please correct it, it should be superscript.

These observations have already been addressed in the manuscript.

2.4 Differentiation and transfection of monocyte-derived macrophages 113

Lines 140 and 141; Subsequently, the FITC-conjugated anti-human IgG secondary antibody (Jakson Immuno R® ), diluted in 1………….PBS with 3% skimmed milk powder, missing unit must be added.

Writing units and numbers should be standardized in the whole article. Some different styles can be found at line 95 (4°C) and line 108 (t 36 °C).

These observations have already been addressed in the manuscript.

  1. Results

Table 1, titled Clinical characteristics of the patients could be improved by minimizing or removing the spaces between the rows.

These observations have already been addressed in the manuscript.

3.1 Study population

A total of 14 patients (6 females and 8 males), 9 from Yucatan and 5 from Baja California, Mexico, were positive for RfelisR. typhi, and R. rickettsii, confirmed by PCR. The healthy subjects (how many) were of the same age, region, and ancestry. None of the patients and healthy subjects were under treatment prior to study enrollment (Table 1).

-          Author should mention the number of healthy samples used in this study. Also should mention the numbers of males and females in both patients and healthy samples.

-          Also authors have to compare the differences in the results between males and females.

  •          As well as the differences in the percentage of macrophages (CD14+) expressing CD40+, CD80+, HLA-I and HLA-II in both males and females should be statistically compared

Aswer:

The size of the Population sample (N=14) belonging to the group of infected patients and confirmed with the PCR test is relatively small. When carrying out the inferential analysis of said group but stratified by gender (6 women and 8 men), the inferential analysis is more detailed, with possible structural bias in the statistic to be used, which, in this case, would be the student's T test for independent samples, where most likely there are no statistically significant differences in the averages obtained in both groups

  1. Discussion

-          The first sentence is too long and should be divided and rewritten correctly with references.

The social impact of the development of a vaccine against Rickettsia is important since there are currently no commercial methods for the diagnosis of rickettsiosis in the acute phase, which is found mainly in vulnerable communities away from specialized health services where patients are in critical conditions and, if they don’t find access to treatments, eventually become fatal cases (………..???).

  •          At the end of the discussion part, authors have to add the limitation of this study as small sample size and so on……

These observations have already been addressed in the manuscript.

  1. Conclusion: could be improved.

References:

Well written according to the journal instructions.

Need revision and should be standardized, I have seen some references without page numbers, please recheck all the references. For example; reference 7, and 15.

These observations have already been addressed in the manuscript.

Reviewer 2 Report

Dear Athors

The manuscript is well written, overall methods described in this paper is good and presented very well,  however there are some deficiencies and questions which can be addressed in a revised version. 

1. It will be useful to know about the viability of T cells after priming with macrophages pulsed with transfected with the plasmid pVAX1-OmpB24 pVax1 over time and also to know access some of the T-cell exhaustion markers.

2. T cell recognize antigen are MHC/HLA dependent, are each macrophages/T cells same donor? It will be useful to know the donors PBMCs HLA-type information be included in the paper.

With regards

Author Response

Response to the reviewer

  1. It will be useful to know about the viability of T cells after priming with macrophages pulsed with transfected with the plasmid pVAX1-OmpB24 pVax1 over time and also to know access some of the T-cell exhaustion markers.

This observation has already been addressed in the manuscript.

  1. T cell recognize antigen are MHC/HLA dependent, are each macrophages/T cells same donor? It will be useful to know the donors PBMCs HLA-type information be included in the paper.

Answer:

As described in the material and methods section, all the cultures are autologous, that is, the macrophages and T cells are from the same donor. On the other hand, we recognize the relevance of the HLA type in the donors, as you mention, however, we do not have detailed information on the specific HLA type, since the objectives of the work were focused on a general analysis of the cellular phenotype. . The only data we have is related to the antibodies used in this study, being: HLA-A/B/C (Biolegend cat. 311413) for HLA-I; and HLA-DR (Biolegend cat. 307620) for HLA-II.

Reviewer 3 Report

The manuscript presented the evaluation of vaccination strategy against rickettsiosis in an autologous system with macrophages transfected with the vaccine candidate pVAX1-OmpB24. Before the paper could be accepted for publication in Vaccines, the authors need make revisions as follow:

1. Centrifugal force should be unified and changed with --- ╳g.

2. Statistical analysis methods should be supplemented, and data should be statistical analyzed in Fig.3, 4 and Table 2, 3.

3. Tables should be normalized and unified with three lines table.

4. Fig. 1 was Agarose gel electrophoretogram? How to analyze the quantitative results?

5. “Image observed at 40x magnification” in Figure 2 legend, was wrong.

6. The vaccine candidate pVAX1-OmpB24 was plasmid? Please gave background about it.

Author Response

Response to the reviewer:

  1. Centrifugal force should be unified and changed with --- â•³g.
  2. Statistical analysis methods should be supplemented, and data should be statistical analyzed in Fig.3, 4 and Table 2, 3.
  3. Tables should be normalized and unified with three lines table.
  4. Fig. 1 was Agarose gel electrophoretogram? How to analyze the quantitative results?
  5. “Image observed at 40x magnification” in Figure 2 legend, was wrong.
  6. The vaccine candidate pVAX1-OmpB24 was plasmid? Please gave background about it.

Answer:

All observations were addressed and incorporated into the manuscript.